



# In situ airborne measurements of atmospheric parameters and airborne sea surface properties related to offshore wind parks in the German Bight during the project X-Wakes

Astrid Lampert[1], Rudolf Hankers[1], Thomas Feuerle[1], Thomas Rausch[1], Matthias Cremer[1],
Maik Angermann[1], Mark Bitter[1], Jonas Füllgraf[1], Helmut Schulz[1], Ulf Bestmann[1], and Konrad Bärfuss[1]

[1]TU Braunschweig, Institute of Flight Guidance, Hermann-Blenk-Str. 27, 38108 Braunschweig, Germany

**Correspondence:** Astrid Lampert (Astrid.Lampert@tu-braunschweig.de)

**Abstract.** Between 14 March 2020 and 11 September 2021, meteorological measurement flights were conducted above the German Bight in the framework of the project X-Wakes. The scope of the measurements was to study the transition of the wind field and atmospheric stability from the coast to the sea, to study the interaction of wind park wakes, and to study the large-scale modification of the marine atmospheric boundary layer by the presence of wind parks. In total 49 measurement flights were

performed with the research aircraft Dornier 128 of the Technische Universität (TU) Braunschweig during different seasons and different stability conditions. Seven of the flights in the time period from 24 to 30 July 2021 were coordinated with a second research aircraft, the Cessna F406 of TU Braunschweig. The instrumentation of both aircraft consisted of a nose boom with sensors for measuring the wind vector, temperature and humidity, and additionally a surface temperature sensor. The Dornier 128 was further equipped with a laser scanner for deriving sea state properties and two downward looking cameras in the visible

and infrared wavelength range. The Cessna F406 was additionally equipped with shortwave and longwave broadband radiation sensors for measuring upward and downward solar and terrestrial radiation. A detailed overview of the aircraft, sensors, data post-processing and flight patterns is provided here. Further, averaged profiles of atmospheric parameters illustrate the range of conditions. The potential use of the data set has been shown already by first publications. The data of both aircraft are publicly available in the world data centre PANGAEA: https://doi.pangaea.de/10.1594/PANGAEA.955382 (Rausch et al., 2023).

## 1   Introduction

Renewable power from wind turbines already plays a major and increasing role in current and future worldwide energy supply (Veers et al., 2019). Despite the forecast of drastic declines in wind energy cost (Wiser et al., 2021), all phenomena significantly reducing the energy yield have to be considered carefully for the operation of current and the planning of future wind parks (Akhtar et al., 2021), and included in economic efficiency calculations (Lundquist et al., 2019). Due to spatial limitations

and for sharing expensive infrastructure, offshore wind energy converters are mostly arranged as wind parks containing up to 100 individual turbines, and as large clusters of the dimensions of several 10 km to 100 km combining different wind parks. Ideas for using floating platforms or different foundation mechanisms at larger water depth have been developed (Manzano-Agugliaro et al., 2020), and wind measurements at first floating wind parks are currently investigated (Angelou et al., 2023).





Wind parks strongly interact with the atmosphere, and induce changes to the wind resource, e.g. Akhtar et al. (2021). Wind
park wakes reduce the wind speed and increase turbulence in downwind areas (Pettas et al., 2021; Syed et al., 2023). The
wake recovery process can take several tens of kilometers (Christiansen and Hasager, 2005; Li and Lehner, 2013), influencing
downwind located wind parks, e.g. Nygaard and Hansen (2016); Cañadillas et al. (2020). However, this strongly depends on
atmospheric conditions, in particular on the presence and altitude of a temperature inversion (Platis et al., 2020, 2022). Wakes
of large wind parks reduce the available inflow wind speed and reduce the power output of subsequent downstream wind parks
(El-Asha et al., 2017; Schneemann et al., 2020). Further, the vertical exchange is increased, which results in enhanced latent
heat fluxes (Platis et al., 2023). The wake effects are superposed by spatial variability of the wind field, in particular for wind
from land to sea with sudden changes in surface properties at the coastal transition (Djath et al., 2022).

As wind speed is one of the main drivers for sea state development, the reduced wind speed in wakes also results in modified
wave spectra (Bärfuss et al., 2021), sediment transport (Rivier et al., 2016), ocean stratification and upwelling (Paskyabi, 2015),
ocean circulation (Broström, 2008), and a reduction of significant wave height (Ponce de León et al., 2011; Fischereit et al.,
2022).

Therefore, it is of crucial importance to improve the knowledge about atmospheric processes and impacts related to wind energy
extraction and dynamics on multiple scales, from individual turbine blades to large wind park clusters, to assess manipulation
measures, and to train qualified personnel with interdisciplinary understanding for future decision making and site selection
(Spyridonidou and Vagiona, 2020).

Airborne measurements can be used as flexible tools for directly measuring the offshore wind field and surface properties, and
modifications induced by wind parks on large scales (Platis et al., 2018). In the framework of the project WIPAFF (WInd PArk
Far Field, Platis et al., 2020), an extensive airborne data set has been acquired to demonstrate the existence and extent of wind
park wakes under stable conditions (Bärfuss et al., 2019; Lampert et al., 2020). Since the acquisition of this first comprehensive
data set in 2016/2017, new wind park clusters have been built in the German sector of the North Sea. Therefore, the project
X-Wakes acquired additional airborne data sets in 2020/2021 with the aim to investigate the following phenomena:

- interaction of wakes of different wind parks

- large scale effects of wakes on downstream wind parks

- global blockage effect modifying the flow field upstream of wind parks

- coastal transition of wind field and stability

One key challenge of interpreting airborne measurements is the fact that the wind field is both temporally and spatially variable,
and the assumption of steady conditions is not valid for distances of 100 km and time scales of several hours. Therefore, two
aircraft were deployed simultaneously for several cases, one measuring the flow field upstream of a wind park or wind park
cluster, the other measuring the downstream flow field. The same was done for studying the coastal effect with two aircraft.
The aim of this article is to provide an introduction to the airborne measurements published in Rausch et al. (2023). The
structure is as following: Section 2 presents the two aircraft of TU Braunschweig used for the measurement flights. Section 3



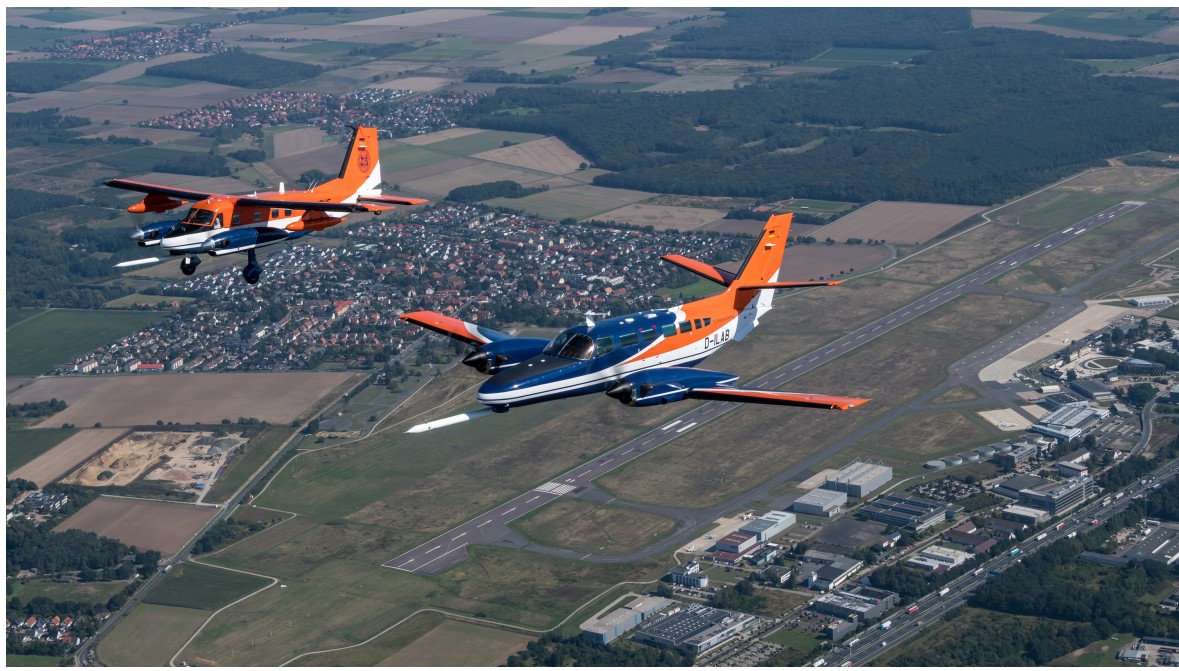

**Figure 1.** The research aircraft Dornier 128 D-IBUF (left) and Cessna F406 D-ILAB (right) with striking painting for good visibility and the characteristic nose boom for meteorological measurements flying above their home base, the airport Braunschweig. Florian Szczepanek, AviationMedia.com

shows the instrumentation of the two aircraft. Section 4 explains the different flight patterns applied to study the phenomena mentioned above, and references to detailed studies already performed with the data set. Section 5 illustrates the atmospheric conditions during the measurement flights. Section 6 presents exemplary measurements of wind park wakes, blockage effect, coastal effect, the wind field modifications above wind parks, radiation measurements for cloud identification, and small-scale changes of sea surface. Finally, Sect. 7 provides a conclusion.

## 2   Research aircraft

Two research aircraft were used for the investigation of wind park wakes and modifications of the atmospheric boundary layer in the vicinity of offshore wind parks during the process of replacing the measurement aircraft Dornier 128 of TU Braunschweig by the successor aircraft Cessna F406 (Fig. 1). In 2021, the last meteorological measurements campaigns of the Dornier 128 and the first campaigns of the Cessna F406 were performed in parallel.

### 2.1   Dornier 128

The Dornier 128 with call sign D-IBUF (Do128-6) is a twin-engine propeller aircraft which has been deployed by TU Braunschweig for atmospheric research from 1986 until 2021. It was operated at an air speed of 60-65 m s$^{-1}$. The aircraft is described





in detail in Corsmeier et al. (2001). An overview of meteorological measurement campaigns with this research aircraft is given in Lampert et al. (2020).

## 2.2   Cessna F406

The Cessna F406 with call sign D-ILAB is a twin-engine propeller aircraft which has been chosen as successor for the successful aircraft Dornier 128. It has similar specific properties which are required for operation in the atmospheric boundary

layer. The non-pressurized cabin allows to install equipment in the fuselage looking through openings in the roof and on the bottom. It can fly at a low cruising speed of $70\,\mathrm{m\,s^{-1}}$ for high resolution measurements. The two engines enable operations at low altitudes, as required for the flights at hub height above the North Sea. The aircraft provides space for two pilots and two to three operators, depending on the installed measurement equipment in the cabin.

The state of the art data acquisition system consists of a core computer with several interfaces (like different digital and analog

input and output signals, including the format ARINC429, serial and Ethernet ports, etc.). This computer receives the sensor data in real time, provides time stamps to the data packages and stores the raw data items on an internal hard disk. It provides the raw data via Ethernet to a second computer, which makes real time calculations for e.g. wind speed in all three dimensions. The second computer performs unit conversions and applies calibration and correction factors where required. At the time of the campaign, it provided a text-based user interface with tables and columns for several raw and calculated parameters. Now

it also includes a graphical user interface with maps, primary flight display, system surveillance and protocol. The pre-selected parameters are recorded on a removable hard disk which allows for quick data transfers after a measurement flight. The raw data on the first computer remain in the aircraft as backup and are overwritten after a couple of flights. This approach ensures reliable data handling and data security. A third computer is used to provide online graphics and quicklooks of different parameters since the beginning of the recording. This feature displays time series and vertical profiles and can be used for

onboard decision making, for example, to determine the most relevant flight altitude for a specific missions. The overall layout of the data acquisition and recording system allows to flexibly integrate additional sensors for synchronous recording. The user interface for monitoring of the flight and mission parameters as well as an interaction with the system is installed on two different mission consoles. In addition, a wireless access point is implemented to stream the real time data inside the cabin, which allows more scientists or observers to monitor the acquired parameters on a mobile device as well.

Figure 2 shows a front view of the aircraft with the nose boom. Another sensor below the aircraft nose next to the noseboom is a forward looking camera. The location of the measurement head in front of the aircraft is a compromise between structural requirements on the aircraft frame and distance in front of the aircraft to perform measurements in the undisturbed air flow.

The data acquisition system can record so-called "events", i.e. specific points in time when one of the cockpit crew members or one of the scientists/observers has detected a noticeable event during the flight and triggers the "event button". Such an event

increases an internal counter and writes a short protocol note into the logfile, complemented by a photo of the forward looking camera. This feature is very useful during the process of data evaluation.




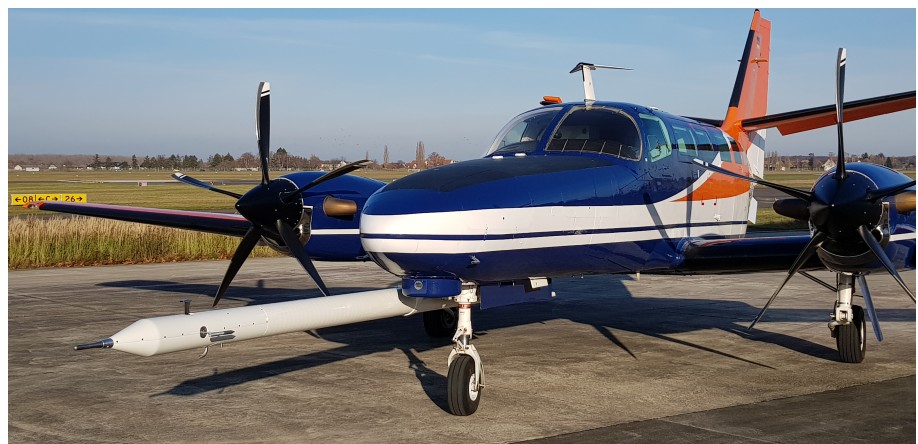

**Figure 2.** The research aircraft Cessna F406 D-ILAB. Thomas Feuerle

## 3 Sensors and data processing

The instrumentation and data processing of the Dornier 128 have been described in detail in Lampert et al. (2020). Therefore, the instrumentation of the Dornier 128 is only shortly summarized. The instrumentation of the Cessna F406 is presented in more detail, in particular if it is different from the sensors onboard of the Dornier 128.

### 3.1 Temperature

The concept of using two complementary sensors for temperature has been implemented for both aircraft. In the data processing, the data from a highly accurate temperature sensor 102DB1AG (Rosemount, USA) and a fast temperature sensor 102E4AL (Rosemount, USA) can be combined or used for quality check. Further, the effect of self-heating induced by the air speed is taken into account in the data processing (Stickney et al., 1994; Bärfuss et al., 2018).

### 3.2 Humidity

Humidity is measured with three sensors based on different measurement principles: In the nose boom of the Dornier 128, there is a dew point mirror TP 3-S (Meteolabor, Switzerland), a capacitive sensor Humicap H233 (Vaisala, Finland) and a Lyman-Alpha sensor L-6/HMS-2 (Buck Research, USA). The dew point mirror provides the most precise measurements at a temporal resolution of 1 Hz. The Lyman-Alpha, based on optical absorption in the ultraviolet wavelength range, delivers data with a high temporal resolution of 100 Hz, but a drifting signal.

In the nose boom of the Cessna F406, there are a dew point mirror and capacitive sensor of the same type, and the optical sensor KH-20 (Campbell Scientific, USA) based on absorption in the ultraviolet wavelength range for fast water vapour fluctuations.





## 3.3 Wind vector and pressure

Determining the wind vector is based on the principle of subtracting the air speed vector from the ground speed vector. The ground speed vector is derived from the high precision measurements of the aircraft position and attitude using an integrated system of Inertial Measurement Unit (IMU) and Global Navigation Satellite System (GNSS). For the air speed vector, measuring the airflow angles, angle of attack and sideslip, is required. This is done via a five-hole probe at the front of the nose boom, and needs fusing the data with high precision attitude data (Lenschow, 1972). The static and dynamic pressure as well as the pressure differences occurring at the five-hole probe (Rosemount, USA) are measured with pressure transducers of Setra, USA. The pressure transducers are located right behind the probe to minimize the length of the required tubes. While on board the Dornier 128 the inertial data is measured with the IMU iNAV-RQH-1003 (iMAR, Germany) and the GNSS data with an external GNSS receiver OEM6 (NovAtel, Canada), on board the Cessna F406 the GNSS receiver OEM6 (NovAtel, Canada) is already integrated into the successor IMU iNAT-RQT-4001 (iMAR, Germany). Accuracy and resolution of the systems used are compared in Table 1. The accuracy of the three wind speed components is better than $0.2 \, \mathrm{m \, s^{-1}}$.

**Table 1.** Technical data of the Inertial Measurement Units iNAV-RQH-1003 (Dornier 128) and iNAT-RQT-4001 (Cessna F406)

| Aircraft | | Dornier 128 D-IBUF | Cessna F406 D-ILAB |
|---|---|---|---|
| Inertial Navigation System | | iNAV-RQH-1003 | iNAT-RQT-4001 |
| Range | Gyroscopes | $\pm$ 400 °/s | $\pm$ 395 °/s |
| | Accelerometers | $\pm$ 20 g | $\pm$ 20 g |
| Bias Stability | Gyroscopes | 0.002 °/h | < 0.001 °/h |
| | Accelerometers | < 10 $\mu$g | < 12 $\mu$g |
| Resolution | Gyroscopes | 0.0003 ° | 0.00033° |
| | Accelerometers | < 5 $\mu$g | < 5 $\mu$g |
| Scale Error | Gyroscopes | < 5 ppm | < 15 ppm |
| | Accelerometers | < 100 ppm | < 100 ppm |
| Linearity Error | Gyroscopes | < 5 ppm | < 10 ppm |
| | Accelerometers | < 20 $\mu$g/g$^2$ | < 30 $\mu$g/g$^2$ |
| Data Rate | | 300 Hz | 400 Hz |
| True Heading | | < 0.04 °/sec(lat) | < 0.028 °/sec(lat) free inertial < 0.01° with GNSS |
| Attitude Accuracy | | < 0.01° | < 0.025° free inertial < 0.01° with GNSS |
| Position Accuracy | | < 0.8 nm/h | < 0.8 nm/h free inertial < 1.6 m with GNSS |
| GNSS Receiver | | NovAtel OEM6 (external) | NovAtel OEM6 (internal) |

### 3.4 Surface temperature

The surface temperature of both aircraft is measured with an infrared sensor (KT15.82D in the Dornier 128 and KT19.85 in the Cessna F406, both of former Heimann, now Heitronics, Germany). As the infrared radiation passes through the atmosphere between the surface and the aircraft, with an unknown concentration of humidity, particles or other factors that potentially influence the infrared radiation, the measured surface temperature depends on the flight altitude with a linear factor of typically between 0.1 and 0.15 K/100 m.

### 3.5 Sea surface deflection

The laser scanner VZ-1000 (Riegl, Austria) is included onboard the Dornier 128 to measure sea state properties through an opening in the fuselage. With this setup it is possible to measure the significant wave height (Bärfuss et al., 2020) and derive wave spectra (Bärfuss et al., 2021) along the flight path. The laser scanner covers a line of sight range of up to 450 m.

### 3.6 Cameras

In the Dornier 128, two downward looking cameras are integrated: the MV1-D1312-G2 (Photonfocus, Switzerland) for the visible wavelength range, and the A35SC (FLIR, Germany) for the infrared wavelength range. In the Cessna F406, nadir looking cameras were not yet installed.

### 3.7 Radiation

Only in the Cessna F406, sensors for measuring broadband upward and downward solar and terrestrial irradiance are included: two pyranometers CMP22 (Kipp and Zonen, The Netherlands) for measuring solar irradiance in the wavelength range 210-3600 nm, and two pyrgeometers CGR4 (Kipp and Zonen, The Netherlands) for measuring terrestrial irradiance in the wavelength range 4.5-42 $\mu$m. The sensors' response time is given as <1.7 s for the solar radiation and <18 s for the terrestrial radiation, which is quite low for the high ground speed of the aircraft and cloud structures typically varying on the scale of few 100 m to 1 km for shallow convective clouds (Fast et al., 2019). Radiation measurements have been included to complement the atmospheric measurements and as an indicator to identify cloudiness.

## 4 Flight planning and flight patterns

Generally the flights had the aim to provide direct in-situ information on modifications of the wind field and the structure of the atmospheric boundary layer related to offshore wind energy. In particular the interaction of wakes of different neighbouring wind parks and wind park clusters was investigated in order to quantify these effects, understand processes, validate simulations and enable realistic forecasts of wake effects for future wind park scenarios. For studying wakes and wake interaction, the flight pattern MEANDER (Sect. 4.1) was applied. The wind speed reduction in wakes can be in the order of up to 30% (Cañadillas et al., 2020).



Further, the erection of large wind park clusters provides an obstacle for the flow, with the consequence that the air tends to flow partly around and above the wind parks instead of passing straight through the wind turbines. This so-called blockage effect was investigated with a particular pattern (BLOCKING) upstream of the wind park clusters (Sect. 4.2). The expected order of magnitude of the effect is much lower, in the range of up to 4% (Schneemann et al., 2021).

The phenomenon of the coastal transition was another target of the measurement flights. The coastal effect means that the abrupt changes of the surface properties at the coast, with lower surface roughness and different heat capacity, lead to modifications of the wind field and the thermal stratification (Schulz-Stellenfleth et al., 2022), which is particularly challenging to include in simulations (Siedersleben et al., 2018). The increase or decrease of wind speed due to synoptic scale changes and the coastal effect can be in the order of several $\mathrm{m\,s^{-1}}$ (e.g., Platis et al., 2018; Djath et al., 2022). The flight pattern COAST (Sect. 4.3)
was applied.

Finally, flights were not only performed upstream or downstream of the wind parks, but also above the wind parks to investigate changes in the boundary layer. This was done with the flight pattern ABOVE (Sect. 4.4).

An overview of all trajectories flown during the 49 X-Wakes flights in the German Bight is provided in Fig. 3. Also the names of the wind park clusters N2, N3 and N4 are indicated there. The flexible flight permission with low-level flights down to 50 ft
(around 15 m) was valid for the German airspace. The flights were mostly performed from the airport Wilhelmshaven with ICAO (Civil Aviation Organisation) code EDWI. When no overnight stays were possible due to pandemic travel restrictions, the measurement flights were done directly from the home base, the airport Braunschweig with ICAO code EDVE. An overview of all flights, date, time, prevailing wind speed and wind direction, wind parks, flight patterns, cloud conditions and simultaneous satellite overpasses is provided in Tab. 2 and Tab. 3.

### 180  4.1  Meander at hub height (MEANDER)

To investigate the interaction of several wind parks or even wind park clusters, the flight pattern MEANDER was applied (Fig. 4a). This means that transects perpendicular to the mean wind direction were flown across the interacting wind park wakes. The distance between the legs was adapted to the atmospheric stability, as very long wakes occur mostly for stable conditions (Cañadillas et al., 2020). For stable conditions, the spacing between legs was typically 10 km. If possible, flight legs were
185 performed at hub height upstream, between wind parks and downstream. A safety distance of 500 m was kept to the closest row of wind turbines. Further, vertical profiles were performed at different locations and times from an altitude of 15 m up to 1 km, if permitted by clouds, to document atmospheric stability and wind shear. For the occasions of two aircraft, the second aircraft performed flight legs upstream of the wind park clusters, or above the wind parks. The MEANDER flight pattern was deployed during 17 out of the 49 flights. Results using the MEANDER flight patterns have been published in Cañadillas et al.
(2022) and zum Berge et al. (2024).

### 4.2  Blockage effect (BLOCKING)

To investigate the flow deflection around wind parks, flights were performed directly upstream of wind parks (Fig. 4b). The flight permission allowed to approach to a minimum distance of 500 m to the closest wind turbines. As the effect is expected

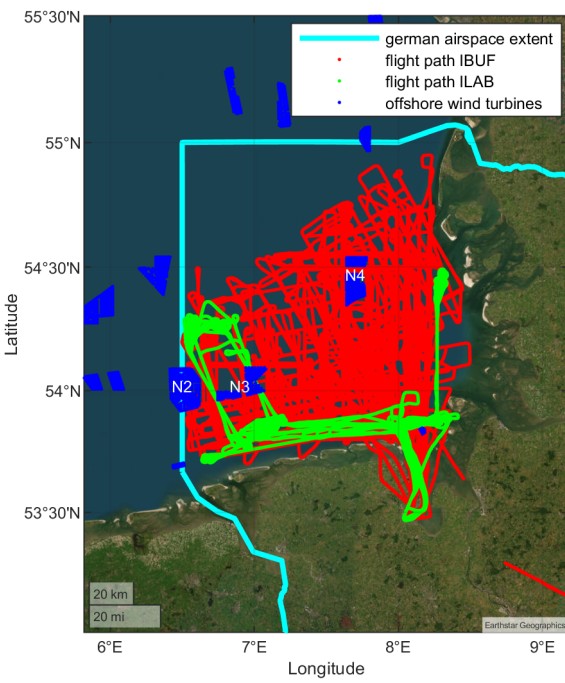

**Figure 3.** Overview of the airborne measurements conducted in the framework of the project X-Wakes. The light blue line indicates the borders of the German airspace for which the flight permission was valid. Wind turbines in operation on the last flight day, 11 September 2021, are displayed in dark blue. All the flight tracks of the Dornier 128 are shown in red, and the flight tracks of the Cessna F406 in green. The names of the wind park clusters N2, N3 and N4 are indicated in white letters,

to be only small, in the range of few %, flights were performed as close to the wind parks as possible, with upstream distances
between 500 m and 2 km and distances between legs from 100 m to 500 m. The BLOCKING flight pattern was deployed during
16 out of the 49 flights. An example of the measurements of the blocking effect is shown in Sect. 6. As the effect is very small,
it is difficult to distinguish it from the spatial and temporal variability, and so far no results using the BLOCKING flight patterns
have been published.

### 4.3 Coastal effect (COAST)

To investigate changes induced by coastal effects on spatial scales of 50 km, meander patterns were performed along the coast
line and perpendicular to the prevailing wind direction at hub height (which is 90 m for the N4 cluster close to the eastern coast
of the German Bight, and 120 m for the N2 and N3 cluster close to the southern coast of the German Bight). Additionally,
vertical profiles were performed at the beginning and end of each leg to document the changes in stability. For the combined
flights with two aircraft, the Cessna F406 repeated the same legs closest to the coast and the Dornier 128 performed meander



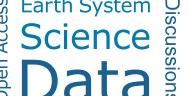

**Table 2.** Overview of the X-Wakes measurements for Flights 1-25 with the research aircraft Dornier 128. The flight patterns are MEAN-DER (M), BLOCKING (B) COAST (C) or ABOVE (A) as indicated in Sect. 4. One short flight was dedicated to comparing sea surface measurements to buoy measurements. The following abbreviations are used for clouds: Cu (cumulus), hum (humilis), Sc (stratocumulus), St (stratus), Ac (altostratus), Ci (cirrus), Cs (cirrostratus), r.s. (rain showers). The overpass times of the satellites Sentinel 1A (S1A) and Sentinel 1B (S1B) are provided as well.

| Flight No. | Date | Time [UTC] | Wind park Cluster | Pattern | Altitude [m] | Wind speed [m s$^{-1}$] | Wind dir ° | Cloud | Satellite |
|---|---|---|---|---|---|---|---|---|---|
| 1 | 14 Mar 2020 | 11:55-15:43 | N3 | B | 120 | 11 | 150 | 3/8 Ac, Ci | S1B, 17:17 |
| 2 | 15 Mar 2020 | 10:01-13:49 | N3 | C | 120 | 14 | 190 | 7/8 Sc | S1A, 17:02 |
| 3 | 17 Mar 2020 | 08:42-12:33 | N4 | B | 90 | 12 | 240 | n.a. | S1A, 05:35 |
| 4 | 21 Mar 2020 | 09:54-14:06 | N4 | C | 90 | 11 | 90 | 1/8 Cu hum | S1B, 17:10 |
| 5 | 21 Apr 2020 | 07:58-11:27 | N4 | C | 90 | 12 | 90 | clear sky | S1B, 05:43 |
| 6 | 21 Apr 2020 | 12:08-15:54 | N4 | C | 90 | 17 | 90 | clear sky | - |
| 7 | 24 Apr 2020 | 07:29-07:31 | N4 | buoy | | 8 | 330 | Sc | S1B, 17:16 |
| 8 | 05 May 2020 | 07:32-11:28 | N4 | B | 90 | 8 | 350 | 5/8 Cu | - |
| 9 | 05 May 2020 | 12:08-16:02 | N4 | B | 90 | 7 | 340 | clear sky | - |
| 10 | 08 May 2020 | 07:29-11:33 | N4 | B | 90, 120 | 7 | 240 | 7/8 Ci | S1B, 05:52 |
| 11 | 08 May 2020 | 12:04-16:05 | N4 | B | 90, 120 | 9 | 240 | 7/8 Ci | S1B, 17:04 |
| 12 | 29 Jun 2020 | 12:58-17:04 | N2, N3, N4 | M | 120 | 15 | 230 | 7/8 Sc, r.s. | S1A, 17:19 |
| 13 | 30 Jun 2020 | 07:48-11:38 | N4 | B | 90 | 17 | 250 | 7/8 Sc, r.s. | - |
| 14 | 30 Jun 2020 | 12:28-16:26 | N4 | B | 90 | 17 | 260 | 5/8 Sc, Ac, Ci, r.s. | S1B, 17:08 |
| 15 | 01 Jul 2020 | 10:36-14:43 | N3 | C | 120 | 10 | 180 | 6/8 Cu, r.s. | S1A, 05:52 |
| 16 | 02 Jul 2020 | 10:54-14:04 | N4 | B | 90 | 3 | 310 | 6/8 Sc, Ci, r.s. | S1B, 05:43 |
| 17 | 03 Jul 2020 | 09:43-13:44 | N2, N3, N4 | M | 120 | 10 | 240 | 8/8 Sc, r.s. | S1A, 05:35 |
| 18 | 13 Jul 2020 | 13:12-17:24 | N4 | B | 90 | 6 | 220 | 3/8 Ac, 7/8 Cs | S1A, 17:02 |
| 19 | 14 Jul 2020 | 12:26-16:05 | N2, N3 | M | 120 | 7 | 270 | 2/8 Cu hum, 7/8 Sc, r.s. | S1B, 05:43 |
| 20 | 15 Jul 2020 | 07:42-11:42 | N4 | B | 90 | 5 | 320 | 2/8 Cu, 6/8 Ac | S1A, 05:35 |
| 21 | 15 Jul 2020 | 12:15-16:01 | N4 | B | 90 | 5 | 300 | 2/8 Cu, 4/8 Ac | - |
| 22 | 23 Jul 2020 | 06:58-11:00 | N3 | C | 120 | 8 | 190 | 8/8 St, r.s. | - |
| 23 | 23 Jul 2020 | 11:28-15:30 | N2, N3, N4 | M | 120 | 10 | 225 | 5/8 Sc, r.s. | S1A, 17:25 |
| 24 | 24 Jul 2020 | 07:01-11:15 | N2, N3 | M | 120 | 10 | 270 | 2/8 Cu | S1B, 17:16 |
| 25 | 27 Jul 2020 | 09:23-13:30 | N3 | C | 90-240 | 10 | 180 | 8/8 As | S1A, 05:40 |

flight legs from the coast to the open water, to be able to separate effects of temporal changes in the inflow conditions and spatial effects induced by the coast.





**Table 3.** Overview of the X-Wakes measurements for Flights 26-49 with the research aircraft Dornier 128. The flight patterns are MEANDER (M), BLOCKING (B) COAST (C) or ABOVE (A) as indicated in Sect. 4. During one short flight, only profile measurements were obtained. The 7 flights with two aircraft in parallel were conducted between 24 July 2021 and 30 July 2021 (numbers printed in bold letters). The same symbols and abbreviations are used as in Tab. 2.

| Flight No. | Date | Time [UTC] | Wind park Cluster | Pattern | Altitude [m] | Wind speed [m s$^{-1}$] | Wind dir ° | Cloud | Satellite |
|---|---|---|---|---|---|---|---|---|---|
| 26 | 28 Jul 2020 | 06:52-10:50 | N2, N3 | M | 120 | 13 | 250 | 3/8 Cu, r.s. | - |
| 27 | 28 Jul 2020 | 11:20-15:15 | N4 | B | 90 | 13 | 250 | 2/8 Cu, 4/8 Ac | - |
| 28 | 23 Sep 2020 | 05:15-09:18 | N3 | C | 120 | 12 | 220 | 5/8 Sc | S1A, 05:55 |
| 29 | 23 Sep 2020 | 10:59-15:03 | N3 | C | 120 | 10 | 210 | 2/8 Sc, 7/8 Ac | S1A, 17:08 |
| 30 | 24 Sep 2020 | 07:07-10:56 | N2, N3, N4 | M | 120 | 17 | 230 | 2/8 Cu | S1B, 05:48 |
| 31 | 24 Sep 2020 | 11:56-12:46 | | profiles | | 10 | 210 | 2/8 Cu | - |
| 32 | 24 Sep 2020 | 13:21-16:35 | N3 | C | 120 | 13 | 190 | 7/8 Sc, r.s. | - |
| 33 | 25 Sep 2020 | 09:45-13:54 | N3 | C | 120 | 10 | 180 | 3/8 Cu, Cs | S1A, 05:40 |
| 34 | 08 Apr 2021 | 07:51-11:53 | N4 | B | 90 | 11 | 260 | 7/8 Sc, r.s. | - |
| 35 | 08 Apr 2021 | 12:21-16:30 | N2, N3, N4 | M | 120 | 15 | 240 | 7/8 Sc | S1A, 17:16 |
| 36 | 12 Apr 2021 | 07:32-11:36 | N4 | B | 90 | 8 | 260 | 3/8 Cu, r.s. | - |
| 37 | 12 Apr 2021 | 12:03-16:03 | N4 | B | 90 | 9 | 260 | 4/8 Cu, r.s. | - |
| 38 | 13 Apr 2021 | 07:30-11:30 | N3 | M | 120 | 11 | 300 | 5/8 Cu, r.s. | - |
| 39 | 13 Apr 2021 | 12:02-16:02 | N3 | M | 120 | 10 | 300 | 5/8 Cu, r.s. | S1A, 17:24 |
| 40 | 14 Apr 2021 | 07:30-11:25 | N3 | M, B | 120 | 7 | 330 | 2/8 Cu, 3/8 Cs, r.s. | - |
| 41 | 14 Apr 2021 | 11:53-15:55 | N3 | M, B | 120 | 6 | 330 | 3/8 Cu, r.s. | S1B, 17:16 |
| **42** | 24 Jul 2021 | 07:46-12:03 | N4 | C | 90 | 7 | 90 | 7/8 Cs | - |
| **43** | 24 Jul 2021 | 12:40-16:05 | N4 | C | 90 | 7 | 70 | 7/8 Cs | S1B, 17:24 |
| **44** | 27 Jul 2021 | 10:09-14:24 | N2, N3, N4 | A,M | 120 | 10 | 240 | 4/8 Cu, Ac, Ci | S1A, 05:48 |
| **45** | 28 Jul 2021 | 08:34-13:00 | N3 | C | 120 | 10 | 200 | 3/8 Cu, Ci, r.s. | S1B, 05:41 |
| **46** | 29 Jul 2021 | 08:36-10:57 | N2, N3 | A,M | 120 | 17 | 240 | 8/8 Sc, r.s. | - |
| **47** | 29 Jul 2021 | 12:40-17:02 | N2, N3, N4 | A,M | 120 | 17 | 245 | 3/8 Cu, r.s. | - |
| **48** | 30 Jul 2021 | 07:37-12:07 | N2, N3, N4 | A,M | 120 | 10 | 240 | 3/8 Cu | S1A, 17:24 |
| 49 | 11 Sep 2021 | 12:51-17:13 | N3, N4 | M | 120 | 9 | 240 | 8/8 St, r.s. | S1A, 17:17 |

The COAST flight pattern was deployed during 14 out of the 49 flights.

Results using the COAST flight patterns have been published in Schulz-Stellenfleth et al. (2022) and Cañadillas et al. (2023).

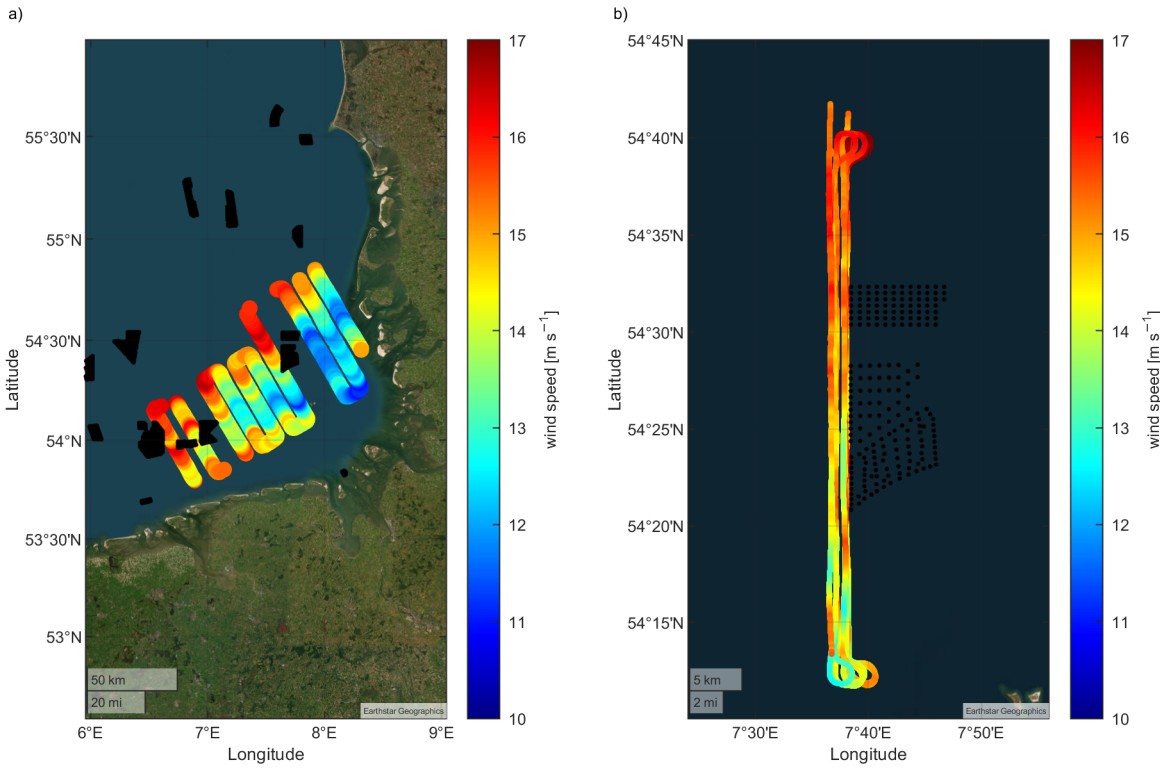

**Figure 4.** a) Example of the MEANDER pattern during Flight 12 on 29 June 2020. The prevailing wind direction was from 230°. b) Example of the BLOCKAGE pattern during Flight 13 on 30 June 2020. The prevailing wind direction was from 250°.

### 4.4 Above wind park (ABOVE)

For the flights with two aircraft dedicated to investigate wakes, the Dornier 128 performed the MEANDER pattern downstream of the wind parks, while the Cessna F406 sampled upwind conditions, and performed additionally flight legs above the wind parks to investigate the effect above rotor height. The flights above the wind parks were done at an altitude of 75 m above the top of the rotor blades, which corresponds to an altitude of 255 m. Flight legs above wind parks were conducted with the Cessna F406 during 4 out of the 7 joint measurement flights. So far no results using the ABOVE flight patterns have been published.

### 5 Atmospheric conditions

To get an overview of the atmospheric conditions during the measurement flights, the following sections illustrate the vertical profiles with the median values, the 25% and 75% percentile and a histogram to indicate the occurrence of the values during



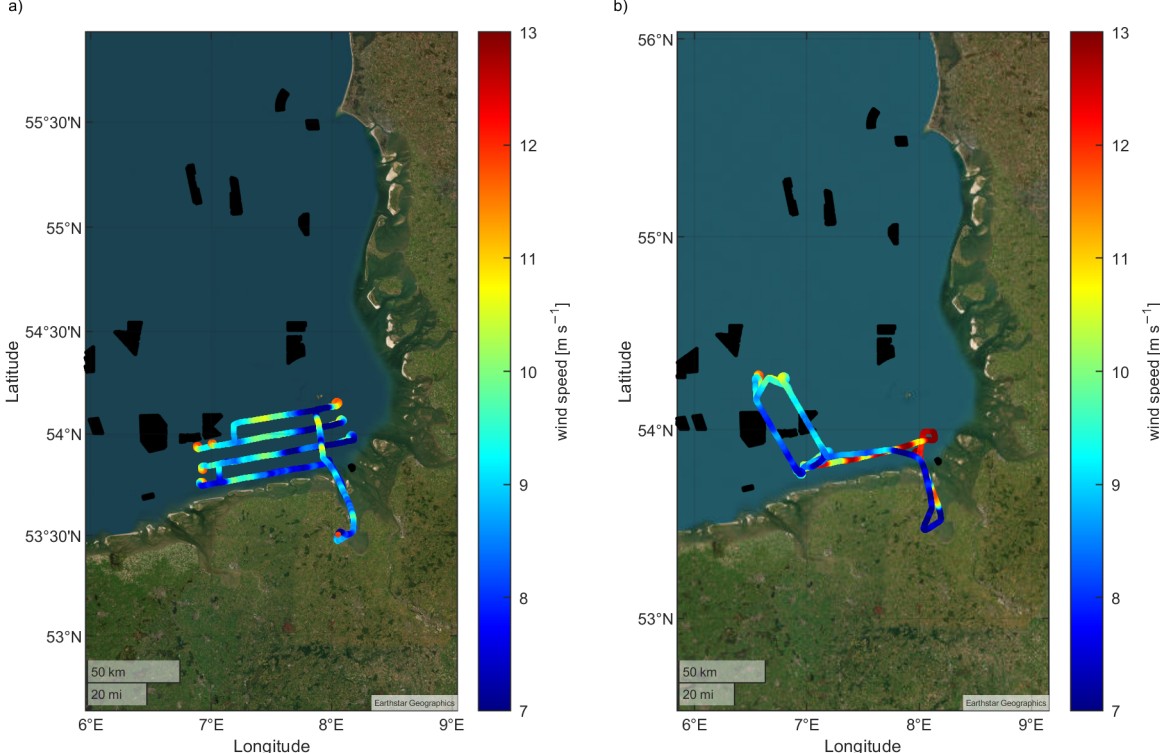

**Figure 5.** a) Example of the COAST pattern during Flight 29 on 23 September 2020. The prevailing wind direction was from $210°$. b) Example of the ABOVE pattern during Flight 48 on 30 July 2021. The prevailing wind direction was from $240°$.

the flights. The measurements are not representative of a climatology, as they were conducted during different seasons, mainly
in spring and summer, and under visual flight conditions (cloud base height above $400\,\mathrm{m}$ for broken or overcast cloud cover, visibility exceeding $5\,\mathrm{km}$).

## 5.1 Temperature

As the flights were performed during different seasons (earliest flight of the year 14 March, latest flight 25 September), the temperature is highly variable between the flights. The temperature near surface was between $4°C$ and $19°C$ (see Fig. 6). Generally,
temperature decreased with altitude. A slight temperature inversion up to an altitude of $50\,\mathrm{m}$ is visible in the mean profile, and on average there is a temperature inversion at an altitude of around $850\,\mathrm{m}$. The whole flights were taken into account, therefore, also temperature values during take-off and landing are included, which may differ significantly from temperature values above the North Sea. The grey scale illustrates that the flights were performed as blocks, and not distributed continuously over time: Lower temperatures, with near-surface air temperature in the range of $4°C$ to $5°C$ occurred during the frequent measurements


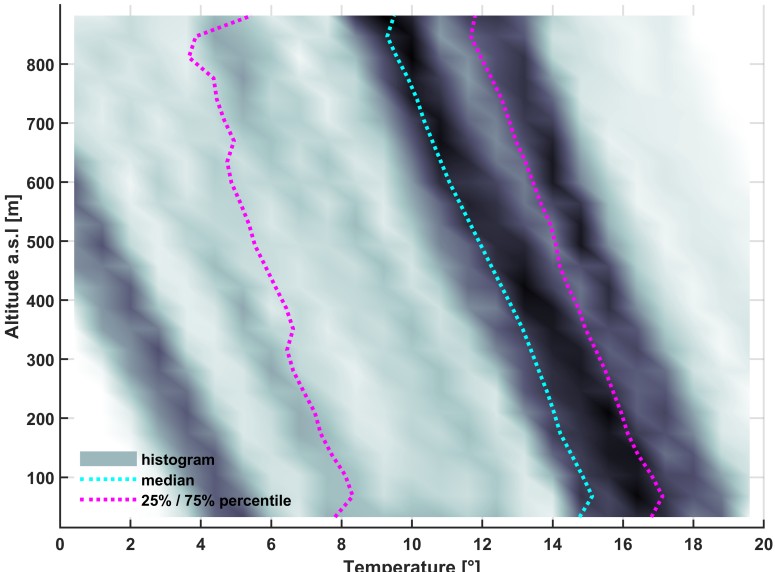

**Figure 6.** Temperature encountered during all measurement flights. The light blue line represents the median of all profiles, the magenta lines indicate the percentiles including 25% and 75% of all values, and the grey scale is a histogram showing the frequency of occurrence of the temperature values.

in early spring (March and April), but most flights were performed during the summer season with relatively warm near-surface air temperature in the range of 15°C to 19°C (see Tab. 2 and Tab. 3). Therefore, the atmospheric conditions during the flights do not represent climatologically relevant statistics.

## 5.2 Stability

Figure 7 shows the distribution of the lapse rate as an indicator of stability for all flights. Values near zero indicate neutral conditions. Most measurements were performed for neutral and slightly stable conditions. For wake measurements and coastal effect, stable conditions were selected on purpose for the measurement flights, as the strongest effects are associated with stable conditions. For investigating the blockage effect, both stable and unstable stratification were probed. During spring and summer, wind directions from land are typically associated with warmer air masses during day than the sea surface, leading to stable stratification. X-Wakes investigated not only the extreme cases of stability, but the interaction of wind park wakes for different stability regimes.

Notable is the enhanced stability at the altitude of the lowest 100 m. On average, there is a strong increase of stability within the lowermost 100 m. This is associated with the altitude of the temperature inversion, see Fig. 6, and has important consequences





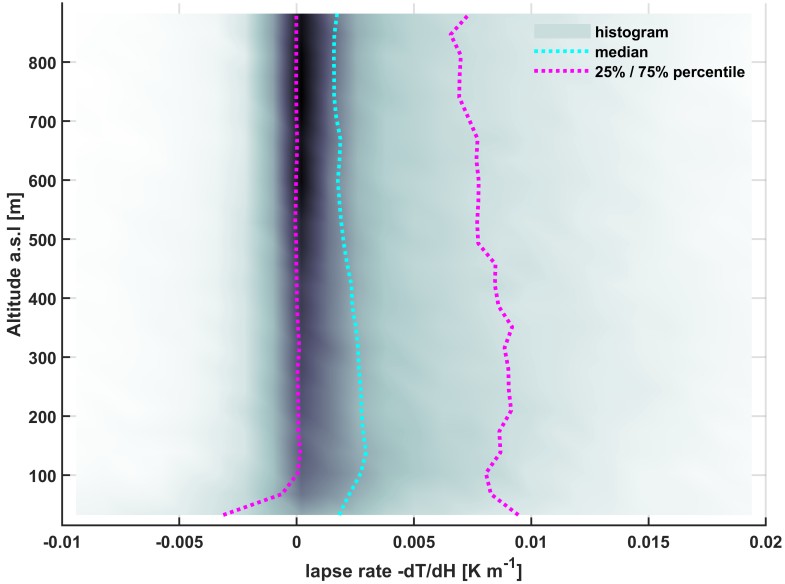

**Figure 7.** Lapse rate as an indicator of atmospheric stability. The light blue line represents the median of all profiles, the magenta lines indicate the percentiles including 25% and 75% of all values, and the grey scale is a histogram showing the probability of occurrence of the lapse rate values.

for the development of the wake, and if there is a warming or cooling effect downstream of the wind park (Siedersleben et al.,
2018).

### 5.3 Wind speed

Figure 8 shows an overview of the wind speed conditions encountered during the measurement flights. The flights were performed under different wind speed conditions. At hub height, the wind speed varied between around $3\,\mathrm{m\,s^{-1}}$ and $17\,\mathrm{m\,s^{-1}}$, but most measurements were performed in the wind speed range between the cut-in speed of $4\,\mathrm{m\,s^{-1}}$, where the wind turbines
start turning, and the rated wind speed of $12\,\mathrm{m\,s^{-1}}$, where the wind turbines reach their maximum power output. In this wind speed range, the power output depends cubically on the wind speed, therefore reductions in wind speed have a direct impact on the energy yield.

### 5.4 Wind direction

Figure 9 shows a wind rose diagram of the wind speed and wind direction that was observed during all measurement flights.
The measurement flights covered average wind directions between 90° and 350° at hub height.

There are several pronounced features: Wind from the South-West sector was frequently associated with a wind speed between

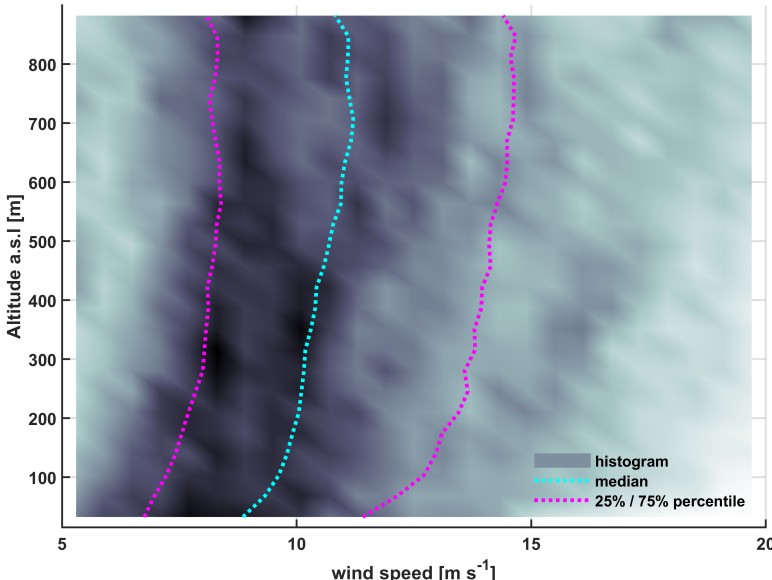

**Figure 8.** Wind speed conditions during all measurement flights. The light blue line represents the median of all profiles, the magenta lines indicate the percentiles including 25% and 75% of all values, and the grey scale is a histogram showing the probability of occurrence of the wind speed values.

$5\,\mathrm{m\,s^{-1}}$ and $15\,\mathrm{m\,s^{-1}}$. This corresponds to the most frequent wind direction for the North Sea (Platis et al., 2018), and is typically associated with stable conditions in spring and summer (Schulz-Stellenfleth et al., 2022). Further, conditions with wind directions from East were measured, which is also typically associated with stable conditions. Several flights were performed during relatively low wind speed from North-West, when typically neutral or unstable atmospheric conditions are expected.

## 5.5 Humidity

Figure 10 shows the frequency of occurrence of different values of relative humidity during all measurement flights. The profiles of relative humidity can be very different depending on atmospheric stability. On average, the relative humidity increases with altitude, which is caused by the decreasing temperature with altitude, as shown in Fig. 6. The decreasing relative humidity within the lowermost $100\,\mathrm{m}$ is in agreement with the increasing temperature in that altitude range.

## 6 Examples of measurements

In this section, a few impressions illustrate what can be done with the data. More and deeper applications with thorough scientific discussions are indicated as references.

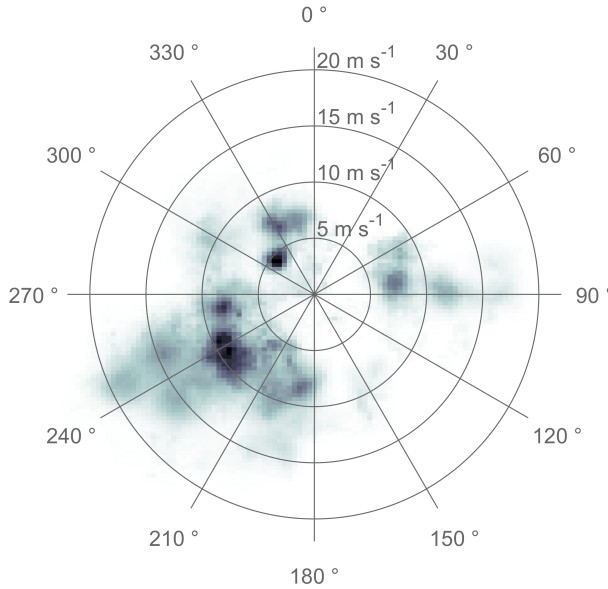

**Figure 9.** Wind rose of the conditions encountered during all measurement flights. The grey scale is a histogram showing the probability of occurrence of the wind speed values in combination with wind direction.

## 6.1 Wind park wakes

Figure 4a shows an example of wakes downstream of several wind parks during Flight 12 in the afternoon on 29 June 2020, potentially superposed by spatial and temporal gradients of the wind speed. Despite the coverage with low-level stratocumulus clouds, the atmospheric conditions were stable, as the temperature of the air masses advected from land was higher than the water surface temperature of the North Sea. The main wind direction at hub height was 230° (South-West), and the mean wind speed was $15\,\mathrm{m\,s^{-1}}$. The changes in the colour from red/orange to blue indicate a reduction of the wind speed to below $13\,\mathrm{m\,s^{-1}}$. This can be observed North-East (downstream) of the wind park clusters N3 and N4. Further, the wind speed is reduced at the South-Eastern edge of the flight pattern, which may be due to the proximity of the coast or downstream of the island of Heligoland. Wind park cluster N2 was partly not in operation on that day according to the flight protocol, which may explain that no pronounced reduction in wind speed is visible downstream.

## 6.2 Blockage effect

Figure 4b shows an example of measurements upstream of the wind park cluster N4 during Flight 13 in the morning on 30 June 2020. The main wind direction at hub height was 250° (West), and the mean wind speed was $17\,\mathrm{m\,s^{-1}}$. A spatial and temporal gradient of the wind field is also obvious here, with higher wind speed towards the North, and changes in wind speed between the individual flight legs, as can be seen by the different colours indicating a wind speed range between $13\,\mathrm{m\,s^{-1}}$ and $15\,\mathrm{m\,s^{-1}}$ in the southern part of the flight, where no influence of the blockage effect is expected. The variability of the wind speed for



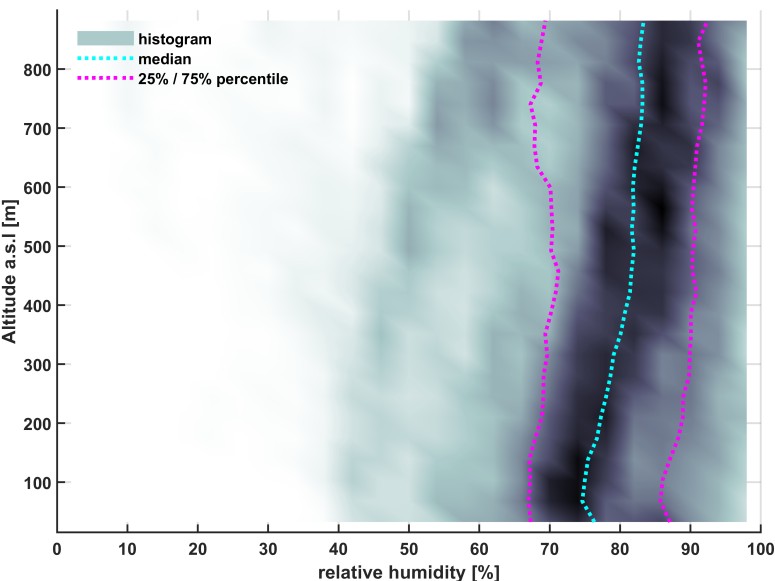

**Figure 10.** Relative humidity encountered during all measurement flights. The light blue line represents the median of all profiles, the magenta lines indicate the percentiles including 25% and 75% of all values, and the grey scale is a histogram showing the probability of occurrence of the relative humidity values.

the time period of one flight leg (around 10 min) and for different flight legs is higher than the expected effect of wind speed reduction of the blockage effect in the range of few %. Therefore, continuously available data sets, e.g. from a scanning wind lidar, are more suitable to identify and quantify the blockage effect (Schneemann et al., 2021).

### 6.3 Coastal effect

Figure 5a illustrates coastal effect, i.e. the spatial variability of the wind field close to and induced by the coast transition from higher to lower surface roughness and from higher surface temperature variability due to lower heat capacity to more constant surface temperatures due to the large heat capacity of the water body. In this case, for wind from South, the wind speed increases with distance from the coast along the wind direction, the so-called fetch length. This is the more frequent case, but also a systematic decrease of wind speed with distance from the coast is possible, as described in Djath et al. (2022). Further, a

high variability of the wind speed along the coast is observed as well. It may be influenced by local orography, with increased wind speed between the different islands, and reduced wind speed downstream, as described in detail in Schulz-Stellenfleth et al. (2022).





### 6.4 Changes in the wind field above wind parks

Figure 5b illustrates the modification of the wind speed during a transect above the wind park N3 at a constant altitude of
250 m. The wind speed is enhanced at the edges of the wind park and reduced directly above. Similar investigations using airborne measurements above wind parks obtained during the project WIPAFF (Bärfuss et al., 2019; Lampert et al., 2020) are presented in Siedersleben et al. (2020) and Syed et al. (2023).

### 6.5 Effect of clouds on radiation

Figure 11 shows an example of the variation of the measured radiation with time when underpassing a cloud. As expected,
the downward solar irradiance decreases under the cloud, visible here as the reduction from around $600\,\mathrm{W\,m^{-2}}$ to $300\,\mathrm{W\,m^{-2}}$. At the same time, the reflected solar irradiance decreases as well, as less radiation reaches the surface. The upward terrestrial irradiance decreases slightly (less than $1\,\mathrm{W\,m^{-2}}$) along the flight leg, probably due to on average slightly decreasing surface temperature (lowest panel of Fig. 11), and independently of the cloud, as the water surface temperature is not changing rapidly with cloud cover. The downward terrestrial irradiance increases slightly (around $1\,\mathrm{W\,m^{-2}}$) due to the enhanced temperature of
the cloud. The unfiltered raw data show spikes in the order of $<0.5\,\mathrm{W\,m^{-2}}$, but the causes have not been identified in more detail yet. Even if the sensor response time provided by the manufacturer is quite high, and despite the hemispheric field of view of the sensors, suitable algorithms can help to estimate the cloud situation during the flights.

### 6.6 Sea surface temperature

Figure 12 shows measured sea surface temperatures in the area of the Jade stream close to the coast for several observation times
within one day. For comparison, sea surface temperature values of the two relevant grid areas of the ERA5 model (Hersbach et al., 2018) are also shown in the diagrams. Depending on the tides and the exact location, the sea surface temperature varies around 2°C during the same day. The difference between ERA5 sea surface temperature and the measured temperature was in the range of 1°C. The gathered data have the potential to deliver contributions for a better understanding of the complex stream and surface temperature situation in coastal areas.

## 7 Conclusions

The X-Wakes flights over the North Sea in 2020/2021 expand the measurements performed during the airborne measurement campaign WIPAFF (Bärfuss et al., 2019; Lampert et al., 2020) in 2016/2017. They serve to investigate the decay of wind park wakes at hub height in the far field. New aspects are

- More wind parks in operation

- New flight patterns BLOCKING and COAST

- Flights also during unstable conditions



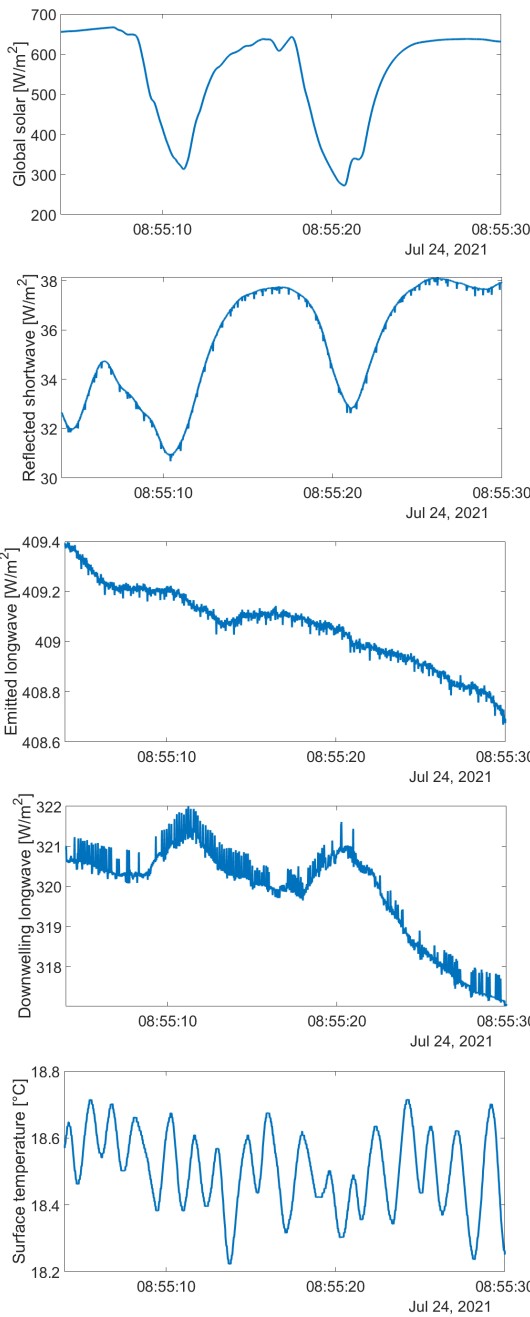

**Figure 11.** Example of the variation in time of the measured irradiance and surface temperature when underpassing a cloud. From top to bottom: Solar downward irradiance, solar upward (reflected) irradiance, upward (emitted) terrestrial irradiance, terrestrial downward irradiance, and surface temperature.

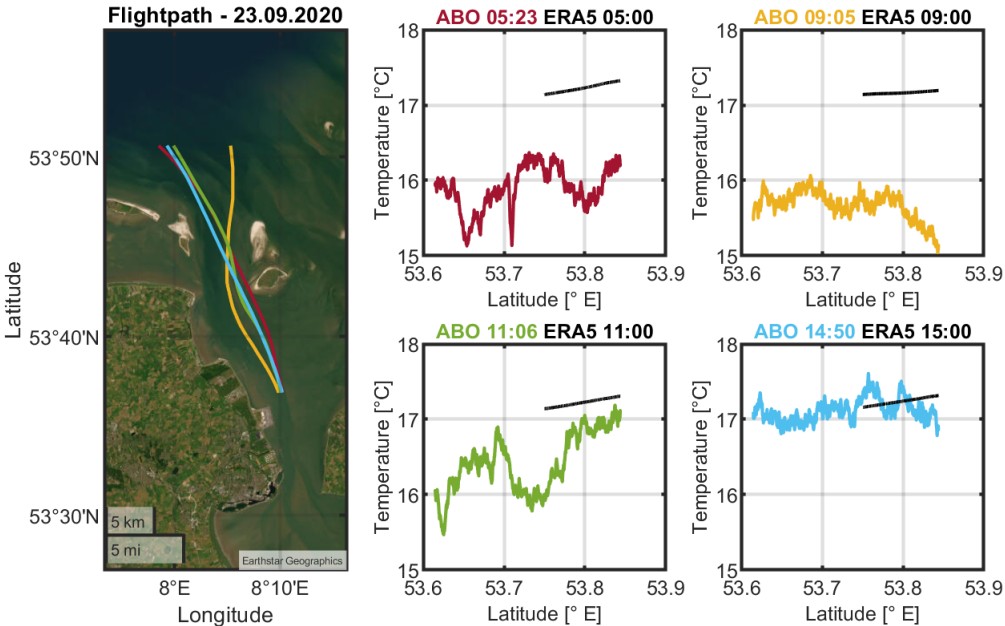

**Figure 12.** Example of sea surface temperature measurements near the coast compared to the two closest points of reanalyses of the model ERA5 for different overpass times, ABO refers to Aircraft Based Observation, times are provided in UTC.

- Some flights with simultaneous observations of two aircraft to separate temporal and spatial variability

- New radiation sensor package on second aircraft

The measurements were embedded in continuous observational programmes like the FINO1 meteorological mast and dedicated wind lidar campaigns, some of which are publicly available as well (Rausch et al., 2023b).

The data have already been used for several investigations within the project X-Wakes, which are referenced in the respective section.

## 8 Data availability

The data are available at PANGAEA: https://doi.pangaea.de/10.1594/PANGAEA.955382 (Rausch et al., 2023). Besides the above-mentioned data on temperatures, humidity, wind, sea surface, and radiation, the dataset contains aircraft position, GPS altitude (WGS84 coordinate system), radar altimeter distance to ground, aircraft velocity (ground speed in north / east / down components), and aircraft attitude. ERA5 reanalysis data are available at Hersbach et al. (2018).

*Video supplement.* A short video with impressions from the research flights is available for illustration purposes.





*Author contributions.*  The authors all contributed directly to creating the data base by performing the measurement flights. A.L., R.H., T.F.,

K.B. and T.R. designed the flight strategy. R.H., T.F., M.C. and M.A. piloted the aircraft. A.L., K.B., M.B., J.F. and T.R. were responsible for the data acquisition. U.B., J.B., M.B. and M.C. developed the data acquisition for the new aircraft. H.S. and M.B. were responsible for airworthiness certification of the sensors and maintenance of the sensors. The manuscript was initiated and drafted by A.L. The figures of the measurement data were provided by K.B. All authors contributed to the writing and editing of the manuscript.

*Competing interests.*  The authors declare no competing interests.

*Acknowledgements.*  The project X-Wakes was funded by the German Federal Ministry of Economic Affairs and Energy (BMWi), now Federal Ministry for Economic Affairs and Climate Action (BMWK) under grant number FKZ 03EE3008B on the basis of a decision by the German Bundestag. The authors would like to thank the partners of the project for the collaboration, and in particular for the fruitful discussions on the preparation of the campaign and the measurement data. The authors would like to thank Johannes Hammel and all other students involved in the analysis of the data.



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
