# Peer review of "In situ airborne measurements of atmospheric parameters and airborne sea surface properties related to offshore wind parks in the German Bight during the project X-Wakes"

_Earth System Science Data, 2024_

## Author Comment (AC1)

**In situ airborne measurements of atmospheric parameters and airborne sea surface properties related to offshore wind parks in the German Bight during the project X-Wakes**

Astrid Lampert, Rudolf Hankers, Thomas Feuerle, Thomas Rausch, Matthias Cremer, Maik Angermann, Mark Bitter, Jonas Füllgraf, Helmut Schulz, Ulf Bestmann, and Konrad Bärfuss

Answers to the Referees

The authors would like to thank the anonymous referee for the valuable comments.

In the following, the comments of the reviewer are answered point by point. The comments are given in *italic*, while the answers are provided in normal letters. Quotations from the new text are given in quotation marks.

Reviewer 1:

*The dataset described in this article seems highly useful and unique, and will be an excellent resource for the wind energy and atmospheric science communities. The comments below are intended to improve the clarity and organization of the article.*

We would like to thank the reviewer for the positive judgement. In the revised version, we take into account all the issues raised in the following.

*I would also recommend making public some of the processing codes that can be used for the data. Codes to separate spatial and temporal variability would be especially useful.*

We are not sure what the reviewer is referring to. In our opinion it does not make sense to publish the code to convert the raw data to the final data set published in PANGAEA. As there was no processing to seperate spatial from temporal variability applied to the published data, there is unfortunately no such code available.

1. *Page 3, line 69: If the Dornier 128 has been in use since 1986, how recently were the sensors calibrated?*

We added in Sect 3: „The temperature, pressure and humidity sensors were calibrated before and after each campaign. The temperature sensors are calibrated using a high-precision resistance decade. All static pressure and differential pressure sensors are calibrated over the respective specified pressure range using two Weston Aerospace DPM7885 absolute pressure transducers as reference. For calibrating the Vaisala Humicap humidity sensor, the sensor head is inserted into a salt chamber containing one of four different saturated salt solutions. The reading given by the probe or transmitter is then adjusted to the humidity value that the specific salt solution generates at that particular temperature. The calibrations described above were carried out on 9 March 2020, 26 March 2021 and 19 June 2021.“

*2. Page 4, lines 96-97: Is vibration corrected for in the measurements?*

We added in the text: „The vibration of the measurement head were measured during the initial flight tests and found to be negligible. So a correction is not required."

*3. Section 3: Throughout this section, it is not always clear which statements apply to both aircraft and which only apply to the Cessna.*

We already specify for most sections the aircraft:
Sect. 3.1: „The concept of using two complementary sensors for temperature has been implemented **for both aircraft**".
Sect. 3.2: „In the nose boom of the **Dornier 128**, …", „In the nose boom of the **Cessna F406**"
Sect. 3.4: „The surface temperature of **both aircraft**"
Sect. 3.5: „… is included onboard the **Dornier 128**"
Sect. 3.6: „In the **Dornier 128**, two downward looking cameras are integrated: the MV1-D1312-G2 (Photonfocus, Switzerland) for the visible wavelength range, and the A35SC (FLIR, Germany) for the infrared wavelength range. In the **Cessna F406**, nadir looking cameras were not yet installed."
Sect. 3.7: „**Only in the Cessna F406**"

We now added:
Sect. 3.3: „Determining the wind vector is based on the principle of subtracting the air speed vector from the ground speed vector for both aircraft."
Sect. 3.5: „During these flights, no laser scanner was available onboard of the Cessna F406."

*4. Section 3: Some of the quantities, such as temperature and humidity, are measured with multiple different sensors. How well do these measurements typically agree? How can discrepancies be reconciled?*

We added in the text: „The method of calculating the best possible result based on a slow, but more accurate, and a fast, but drifting sensor is done by complementary filtering, as described in detail in Bärfuss et al. (2018)"

*5. Page 6, line 124: Please define "sideslip".*

We included in the text a short explanation of the technical terms angle of attack and angle of sideslip:
„For the air speed vector, measuring the airflow angles, the angle of attack (angle between the velocity vector of the aircraft relative to the air and the aircraft longitudinal axis, describing the longitudinal component of the aircraft velocity) and the angle of sideslip (angle between the velocity vector and the projection of the aircraft longitudinal axis, which describes whether there is a lateral component to the aircraft velocity), are required."

*6. Page 6, Table 1: Since many of the gyroscope and accelerometer quantities are in different units, I think the middle part of this table would be easier to interpret if it were laid out as follows:*

| | |
|---|---|
| | Range |
| | Bias stability |
| Gyroscopes | Resolution |
| | Scale error |
| | Linearity error |
| | Range |
| | Bias stability |
| Accelerometers | Resolution |
| | Scale error |
| | Linearity error |

We changed the table as suggested.

*7. Page 7, lines 134-137: This explanation of how surface temperature is measured is not very clear. Can you please add some additional explanation? Also, what is the resulting uncertainty on the measurement?*

We changed the text to: „The surface temperature of both aircraft is measured with an infrared radiation thermometer (KT15.82D in the Dornier 128 and KT19.85 in the Cessna F406, both of former Heimann, now Heitronics, Germany), with an accuracy plusminus 1.2 K at 20°C surface temperature and a temporal resolution of 20 Hz.“
Further, there is already an explanation including uncertainties in the following pragraph: „As the infrared radiation passes through the atmosphere between the surface and the aircraft, with an unknown concentration of humidity, particles or other factors that potentially influence the infrared radiation, the measured surface temperature depends on the flight altitude with a linear factor of typically between 0.1 and 0.15 K/100 m.“

*8. Page 7, line 139: Is sea surface deflection measured by the Cessna too?*

We added in the text:“ During these flights, no laser scanner was available onboard of the Cessna F406.“

*9. Section 4: The text might flow better if the introductory material of this section (i.e., before subsection 4.1) were incorporated into the subsections.*

We re-structured the text as suggested.

*10. Page 8, lines 186-187: How are vertical profiles captured? What is the magnitude of the horizontal translation during these measurements?*

We changed the text to: „In this case the aircraft climbed continuously with a typical vertical speed of 5 m/s and descended again with the same vertical speed, corresponding to a horizontal translation of around 12 km per profile."

*11. Page 9, Figure 3: It is difficult to distinguish the trajectories here. I think it would be better to have a different figure for each flight pattern that can be referenced throughout Section 4. Then Figures 4 and 5 can be moved later in the manuscript to be referenced in Section 6 without having to go back and forth.*

The aim of Fig. 3 is not to distinguish between the different flight patterns, but to provide an impression of the amount of data available. For each subsection with a flight pattern, an example figure is shown. If we show additionally a figure for each flight pattern, it is kind of redundant with the figures showing data along with the flight trajectories. We would prefer to leave the arrangement of the figures to the final typesetting.

*12. Page 9, line 203: Were profiles performed both onshore and offshore?*

We changed the text to: „Additionally, offshore vertical profiles…"

*13.Figures 4 and 5: Add arrows to show the wind direction.*

We added arrows to the plots to indicate wind direction graphically.

*14. Page 12, Figure 4b: The turbines are very hard to see in this figure.*

Thank you for the good hint, we changed the symbols of the wind turbines.

*15. Section 5: What averaging period is used to generate each sample in the histograms?*

There was no plain averaging applied. For all samples, 100 Hz data was filtered by a Chebyshev Type I IIR-filter of order 8 before downsampling to around 7.5 Hz. The lapse rate was additionally median-filtered by a width of 8 samples. These values then are used for the histograms.
We added in the text: "In the illustrations, each sample in the histograms originating from raw 100 Hz was filtered and downsampled to a rate of 7.5 Hz. Additionally, the lapse rate was median filtered with a width of 1 s."

*16. Section 5: While I understand the value of the histograms to give readers a sense of what data is available, I don't think the discussion of the mean temperature and humidity profiles is valuable, as they are averages of different atmospheric stability conditions. It may be more beneficial to separate the profiles by stability.*

With the histrograms we would like to show the overall variability of atmospheric conditions encountered during the different flights. We are not sure about the advantage of plotting separately for different stability, or maybe different wind speed or different wind direction? This leaves open the question which criteria should be applied for the classification of different cases. This seems to be part of  follow-up analyses.

*17. Section 6.3: Why are higher speeds measured at the edges of the flight path rows where the plane turns around?*

We added in the text: „At the edges of the individual legs, vertical profiles were performed to document the vertical distribution of wind, temperature and humidity, with enhanced wind speed for higher altitudes.“

*18. Section 6.5: Make the text and Figure 11 plot axis labels consistent, e.g., "downward solar irradiance" vs. "global solar" and "reflected solar irradiance" vs. "reflected shortwave".*

We harmonised the expressions, and now use the terms „solar downward irradiance, solar upward irradiance, terrestrial upward irradiance, terrestrial downward irradiance“

*19. Page 19, lines 307-308: The decrease in surface temperature is not visible in the figure. Add a trendline or perform a moving average if you want to make this point.*

We added a trend line. We added in the figure caption: „The time series of surface temperature includes a trend line.“ We included in the text: „probably due to on average slightly decreasing surface temperature (lowest panel of Fig. 11, see trend line)“

*20. Page 19, lines 315-316: The comparisons in the top two panels do not look very convincing. Are there any other measurement datasets you can compare against?*

We added in the text:" The agreement between ERA5 sea surface temperature and measured surface temperature was very close for the time of around 14:00 UTC. For other times of day, the difference between ERA5 sea surface temperature and the measured temperature was in the range of 1°C. This indicates that the ERA5 model is not capable of resolving the temporal changes of sea surface temperature associated with tides very well."

*21. Page 19, line 326: If measurements during unstable conditions are a key aspect of this dataset, histograms should be separated by stability (see comment 16).*

We wonder what will be the conclusion of such a classification. We do not even include an indication of stability in Table 2 and 3, as there are different criteria for stability, see e.g.

Platis, A., Hundhausen, M., Lampert, A., Emeis, S., and Bange, J.: The Role of Atmospheric Stability and Turbulence in Offshore Wind-Farm Wakes in the German Bight, Boundary-Layer Meteorology, https://doi.org/10.1007/s10546-021-00668-4, 29 pp., 2021.

And in particular the altitude of the temperature inversion plays a critical role, see e.g.

Siedersleben, S.K., Lundquist, J.K., Platis, A., Bange, J., Bärfuss, K., Lampert, A., Cañadillas, B., Neumann, T., and Emeis, B.: Micrometeorological impacts of offshore wind farm as seen in observations and simulations, Env. Res. Lett., 13, 124012, 2018b.

*22. Page 20, Figure 11: Mark the periods when under the cloud in all the subfigures for clarity.*

We marked the periods of clouds with a grey-coloured bar and added in the figure caption: „The grey bar shows the approximate cloud position.“

*23. Page 21, line 327: If simultaneous observations from two aircraft are important, please show some examples of these results.*

We changed the text to: „For the combined flights with two aircraft (Flight 42 to Flight 48), the Cessna F406 repeated the same legs closest to the coast and the Dornier 128 performed meander flight legs from the coast to the open water, to be able to separate effects of temporal changes in the inflow conditions and spatial effects induced by the coast." Of course we could include a figure showing the trajectories and wind measurements of both aircraft simultaneously, but we would like to leave this for further analyses.

---

## Author Comment (AC2)

**In situ airborne measurements of atmospheric parameters and airborne sea surface properties related to offshore wind parks in the German Bight during the project X-Wakes**

Astrid Lampert, Rudolf Hankers, Thomas Feuerle, Thomas Rausch, Matthias Cremer, Maik Angermann, Mark Bitter, Jonas Füllgraf, Helmut Schulz, Ulf Bestmann, and Konrad Bärfuss

Answers to the Referees

The authors would like to thank the referee Tobias Gerken for the valuable comments.

In the following, the comments of the reviewer are answered point by point. The comments are given in *italic*, while the answers are provided in normal letters. Quotations from the new text are given in quotation marks.

*Reviewer 2 (Tobias Gerken)*

*To move the review process along, I have assigned myself as a reviewer following journal policy.*

*Overall, I find that the manuscript is well-written and that data and data description are useful to the community.*

We would like to thank the reviewer for the positive evaluation.

*I recommend to consider the following issues:*

The points were taken into account as indicated below.

*1. I am not sure I missed this, but could you make sure that you clarify in the text what the reference for above sea level is and how (if at all) this might be different from the actual height above the surface).*

We added in Sect. 3: „The altitude refers to height above mean sea level (WGS84 data from the Global Navigation Satellite System minus geoid height), and the radar altitude provides the height above the surface."

*L 80: Please provide additional calibration and correction information about the instruments. Either here or in other suitable locations*

We added in Sect. 3: „The temperature, pressure and humidity sensors were calibrated before and after each campaign. The temperature sensors are calibrated using a high-precision resistance decade. All static pressure and differential pressure sensors are calibrated over the respective specified pressure range using two Weston Aerospace DPM7885 absolute pressure transducers as reference. For calibrating the Vaisala Humicap humidity sensor, the sensor head is inserted into a salt chamber containing one of four different saturated salt solutions. The reading given by the probe or

transmitter is then adjusted to the humidity value that the specific salt solution generates at that particular temperature. The calibrations described above were carried out on 9 March 2020, 26 March 2021 and 19 June 2021."

*L131: "The accuracy of the three wind speed components is better than 0.2 m s−1" > Please expand on how this is known.*

We changed the text to: „The accuracy of the horizontal wind speed components of the D-IBUF is better than 0.5 m/s and of the vertical wind speed component better than 0.1 m/s (Corsmeier et al., 2001). As pressure measurements are the bottle-neck for wind measurement accuracy, and the D-ILAB deploying the same pressure sensor types as the D-IBUF, no differences in wind speed accuracy between the two aircraft is expected, which was also the observation during calibration flights."

*Table 2 and 3: I suggest o provide measures of variation during flight legs on altitude, wind speed, and wind direction.*

We added in the text: „As a first orientation, the approximate wind direction and wind speed at hub height are provided, which are highly variable in time, in horizontal direction and with altitude."

*Also, specify that satellite overpass time is UTC for completion.*

We added UTC in the tables.

*Figure 4a: Are all of these data from the same altitude or does this include profiles? Please provide additional information in figure legends (or filter for constant altitude). Also, it appears that the large marker size obscures some spatial patterns. I am unsure how this can be avoided practically, but a smaller marker or data gridding might help.*

We changed the marker size as suggested. All data plotted in Figure 4a) is measured around hub height – we clarified this in the caption.

We added in the caption of Figure 4: "The main pattern was flown at hub height, and only data measured at hub height are included in the figures."
We added in the caption of Figure 5: "In contrast to Figure 4, data of the whole flight are included, not only data obtained at hub height."

*Figure 5. Same comments about the turns. It would be good to establish, why wind speeds at the turns appear different than within the legs. Altitude changes or profiles would be a simple explanation.*

We included in the figure caption: „The turns were used to perform vertical profiles, which means that the wind speed was generally higher than during flight legs at constant altitude."

*Figures 6, 7, 8, 10. I am not sure that the histograms in the back are helpful. They certainly should have an associated legend if kept. I would recommend either performing a more meaningful binning*

*or presenting a smaller number of histograms from representative heights in separate subplots. One should also consider conversion to potential temperature.*

The plots should provide a qualitative impression of the atmospheric conditions encountered during the flight to serve as orientation for deploying the data. Future analyses will be based on the PANGAEA data set, and not on such plots. We would therefore prefer to leave the histograms as they are. Potential temperature, or the lapse rate as a derivative of potential temperature, as an important indicator of atmospheric stability, is provided separately in Fig. 7.

*Figure 7: I am a bit confused about the lapse rate. A zero lapse rate in potential temperature would mean neutral. T would be a confusing variable name choice for potential temperature. I suggest clarifying this in the text and also within the figure legend.*

Yes, the lapse rate close to zero indicates neutral conditions. We already state in the text: „Figure 7 shows the distribution of the lapse rate as an indicator of stability for all flights. Values near zero indicate neutral conditions. Most measurements were performed for neutral and slightly stable conditions." As suggested, we changed the variable to theta instead of T.

---

## Author Response (AR2)

**In situ airborne measurements of atmospheric parameters and airborne sea surface properties related to offshore wind parks in the German Bight during the project X-Wakes**

Astrid Lampert, Rudolf Hankers, Thomas Feuerle, Thomas Rausch, Matthias Cremer, Maik Angermann, Mark Bitter, Jonas Füllgraf, Helmut Schulz, Ulf Bestmann, and Konrad Bärfuss

Answers to the Referees:

*Thank you for the revisions to the manuscript that have provided clarifications as requested by the reviewers. Before publication, please add suitable colorbars to the figures with histograms.*

We now included the colour bars for the histograms.